# Zero modes activation to reconcile floppiness, rigidity, and multistability into an all-in-one class of reprogrammable metamaterials

Lei Wu[1] & Damiano Pasini ®[1] ✉

Existing mechanical metamaterials are typically designed to either withstand loads as a stiff structure, shape morph as a floppy mechanism, or trap energy as a multistable matter, distinct behaviours that correspond to three primary classes of macroscopic solids. Their stiffness and stability are sealed permanently into their architecture, mostly remaining immutable post-fabrication due to the invariance of zero modes. Here, we introduce an all-in-one reprogrammable class of Kagome metamaterials that enable the in-situ reprogramming of zero modes to access the apparently conflicting properties of all classes. Through the selective activation of metahinges via self-contact, their architecture can be switched to acquire on-demand rigidity, floppiness, or global multistability, bridging the seemingly uncrossable gap between structures, mechanisms, and multistable matters. We showcase the versatile generalizations of the metahinge and remarkable reprogrammability of zero modes for a range of properties including stiffness, mechanical signal guiding, buckling modes, phonon spectra, and auxeticity, opening a plethora of opportunities for all-in-one materials and devices.

In condensed matter physics, a zero mode (ZM) denotes a particular deformation pattern in a system incurring a fairly low energy cost[1]. ZMs are prevalent in both natural and technological worlds across a wide spectrum of length scales, with examples encompassing but not limited to the shearing of non-viscous fluids and pentamode materials[2,3], the Guest-Hutchinson mode in the Kagome lattice[4,5], and the deployment/retraction of origami-based solar arrays. ZMs are fundamental in sealing the physical properties, e.g., stiffness and stability, of a system, as they represent the most energy-favored pathways of deformation that arise in response to an external stimulus. The intrinsic relation between ZMs and the fundamental physical characteristics contributes to explaining why the notion of ZMs can be used to classify a macroscopic solid into one of three classes, each with its own response that can be diametrically opposite to another (Fig. 1a).

A finite number of ZMs typically defines a mechanism (left in Fig. 1a), which is floppy and hence apt to shape morph but often lacking load-bearing capacity. In contrast, a kinematically determinant structure has no ZMs (right in Fig. 1a), thus showing promising rigidity and robust stability. In between them is a multistable matter (middle in Fig. 1a), which resembles a stable structure around its energy local minima, whereas it behaves akin to an unstable mechanism near its energy local maxima.

Over the past two decades, each individual class of solids (Fig. 1a) has inspired the architected design of mechanical metamaterials attaining a plethora of often strikingly dissimilar mechanical characteristics. The concept of mechanism, for instance, has inspired the design of highly morphable metamaterials[6–10]; the notion of structure has been fundamental to boost the stiffness-to-weight ratio[11,12], and

[1]Department of Mechanical Engineering, McGill University, Montreal, Canada. ✉e-mail: damiano.pasini@mcgill.ca

**Fig. 1 | Role of zero modes in macroscopic solids and activation of a topologically transformable metahinge. a** Primary categories of solids at the macroscopic scale, classified with respect to the number of zero modes. **b** Nearly singular deformation mapping in a solid domain, allowing the formation of a flexible hinge (top row); topology-transformation in a multibody architecture leading to the formation of a contact-induced metahinge (middle and bottom rows). **c** A quarter of the architecture is annotated with key geometry parameters and its deformed configuration. **d** Distinct energy landscapes emerging during activation for given geometric parameters; spots in distinct colors correspond to the activated state on each energy landscape; $E$ is Young's modulus of the elastic constituent material, and $t_0$ is the out-of-plane thickness. **e** Contour plot of $(L − L^*)^2/L^2$ in the design space defined by $\overline{A'B}/\overline{AA'}$ and $\overline{OC}/\overline{AA'}$. **f** Force–displacement relation describing metahinge activation; insets show deactivated (initial) and activated states; the experimental uncertainty domain is obtained by performing identical compression tests at intervals of 10 cycles of activation and deactivation.

that of multistability for self-sustained reconfiguration[13] and elastic energy trapping[14]. Most existing metamaterials, however, lack versatility. Once fabricated with the characteristics of one sole class of solids, they cannot offer those of the others because their architecture is imprinted with an unchangeable number and pattern of ZMs. They behave as either a structure, a mechanism, or a multistable matter, hence inheriting the intrinsic stiffness and stability of the class of solids they belong to. To enable on-the-fly adjustment of stiffness or stability, reprogrammable mechanical metamaterials have been rationally designed to incorporate field-responsive constituents[15–17] or reconfigurable architecture[1,18–22]. A subset of reconfigurable metamaterials has been designed to provide adjustable kinematic determinacy that allows a transition between a structure and a mechanism[1,20,23], offering a remarkable degree of stiffness reprogrammability. Materials exhibiting adjustable geometry incompatibility[19] or local confinement[24,25] can also provide reprogrammability of their stability, enabling a seamless transition between a structure and a multistable matter. Existing reprogrammable metamaterials, however, can only transition between two of the three primary classes of macroscopic solids, falling short in integrating all of them into a single unitary piece of material offering three dissimilar palettes of mechanical properties-rigidity, floppiness, and multistability.

This work introduces an all-in-one class of reprogrammable matter that resolves what has been so far out of reach: crossing the response boundaries between all three classes of macroscopic solids post-fabrication, and enabling the acquisition and switch on site of the traits of each class as desired. This versatility stems from a reversible process entailing the in-situ activation and deactivation of ZMs, which metamorphs the internal architecture to take on the properties of either a compliant mechanism, a rigid structure, or a multistable matter. We demonstrate how to reprogram the ZMs to reconcile the seemingly conflicting mechanical characteristics of all three classes, hence demonstrating multifunctionality across a wide and diverse spectrum of applications. These include - but are not limited to - on-demand mechanical signal guiding, stiffness tuning, selective suppression of buckling modes, mechanical logic operations, tunable phonon spectra, and switchable auxeticity.

## Results

### Activation of metahinge enabling isochoric reconfiguration

The ZMs of a mechanical metamaterial, also known as internal mechanisms[26], are generally realized by connecting relatively bulky bodies via flexible hinges[9,27], hence forming an architecture where the strain energy concentrates within the flexible hinges while the

relatively bulky bodies remain nearly undeformed. To activate/deactivate a ZM entails enabling the emergence or vanishing of flexible hinges. The most intuitive strategy to form a flexible hinge in a conservative system is to squeeze a finite volume of a solid domain into a tiny region $\Omega$, i.e., a flexible hinge (upper part of Fig. 1b), a process that generally leads to a nearly singular deformation gradient $\mathbf{F}$ typically accompanied by excessive material stretching; this requires an abundant amount of energy typically resulting in irreversible plastic deformation or material damage.

An alternative solution to avoid excessive material stretching while still ensuring the formation of a flexible hinge, is to introduce rationally designed perforations into the solid domain. This strategy allows material points on opposite boundaries to engage and form a flexible hinge through local rotation[8]. Fig. 1b (middle) shows a perforated architecture that embodies this principle; it comprises elastic beam-type components (blue) and rigid components (black). In the initial state, its Euler's characteristic is $\chi = -2$. Under the squeeze of a pair of activation forces (red arrows in Fig. 1b), the architecture can undergo moderate local rotation, making the pairing edges (purple) and hinges (green) get close and eventually merge due to self-contact[28,29]. This process of activation transitions the architecture to another topological state characterized by a dissimilar Euler's characteristic, $\chi = -1$. In this activated state, a contact-induced metahinge emerges, empowering the architecture with a rotational ZM about the metahinge. Despite the significant changes in shape and topology, there exists no excessive material stretching in the elastic beam-type components, thus realizing an energy-efficient and reversible activation process.

The process of metahinge activation, which features reflection symmetry, can be monostable or bistable depending on the spacing angle between pairing triangles ($2\alpha$ in Fig. 1b) and the geometry incompatibility ($\beta$ in Fig. 1c). Through a theoretical model (Supplementary Note 1), we study the energy landscapes during activation for varying geometric parameters, i.e., adjusting $\overline{OC}$ and $\overline{A'B}$ to alter $\alpha$ and $\beta$. Three main cases exist. For $\alpha < \beta$, e.g., $\overline{A'B}/\overline{AA'} = 0.7$ and $\overline{OC}/\overline{AA'} = 0.6$, the self-contact between pairing edges occurs prematurely and prevents the architecture from reaching its second stable state, thus resulting in a convex and monostable energy landscape (dashed blue curve in Fig. 1d). For $\alpha > 2\beta$, e.g., $\overline{A'B}/\overline{AA'} = 0.4$ and $\overline{OC}/\overline{AA'} = 2.2$, the architecture is bistable, but its second stable state is a zero-energy state in the absence of self-contact (red spot on the dotted curve in Fig. 1d), meaning that a rotational ZM cannot emerge. Only for $\beta < \alpha < 2\beta$, both bistability and edge contact take place, as shown by two representative energy landscapes plotted with solid curves in Fig. 1d.

An initial version of the above architecture has been leveraged to create one-dimensional mechanical metamaterials with reprogrammable bending stiffness[23] and local resonance[30]. Upon activation, the architecture changes its horizontal length, denoted as $2L^* - 2L$ (Fig. 1b), an outcome that can result in global geometry frustration[31] if this architecture is used as a building block of a tessellated two-dimensional metamaterial. Geometry frustration can substantially alter the volume and external shape of the material, rendering it incompatible with its original boundaries. The architecture previously reported[23,30] fails to address the challenge of geometry frustration, which is resolved here. Upon activation, the horizontal span of our metahinge can be redefined to remain invariant, hence enabling an isochoric reconfiguration process that effectively eliminates undesirable geometry frustration. By tuning $\overline{OC}/\overline{AA'}$ and $\overline{A'B}/\overline{AA'}$, we can effectively alter the length change $L^* - L$ and eventually find conditions that are length preserving. In Fig. 1d, for instance, the yellow energy landscape exhibits $(L^* - L)/L = 0.04$, corresponding to a length change ~27 % that observed in the blue energy landscape, where $(L^* - L)/L = 0.15$. With these insights, we generate a design map where the dimensionless ratio $(L - L^*)^2/L^2$ is plotted with respect to the design variables, $\overline{OC}/\overline{AA'}$ and $\overline{A'B}/\overline{AA'}$. Two curves, $\alpha = \beta$ and $\alpha = 2\beta$ bound the feasible design space. Within the dark blue region in the feasible design space, we ultimately select a pair of values, $\overline{OC}/\overline{AA'} = 2.2$ and $\overline{A'B}/\overline{AA'} = 0.7$ (green spot in Fig. 1e), that guarantee nearly negligible length change $(L - L^*)^2/L^2$ of ~$1.4 \times 10^{-4}$ as well as a well-merged metahinge in the activated state of the as-manufactured specimen. Fig. 1f illustrates the force–displacement relation of the specimen that preserved its length upon activation. A pronounced snap-through instability appears under a pair of squeezing forces, followed by an abrupt and steep rise of the reaction force $F_y$ at the onset of self-contact (purple spot on the solid blue curve in Fig. 1f). The reaction force $F_y$ does not display a significant negative value mainly due to the viscoelasticity of the base material and the untethered loading condition, yet the activated state can be robustly preserved with self-contact. On the other hand, the metahinge is deactivated if a boundary-pulling force is applied to overcome the energy barrier and restore its original shape (Supplementary Movie 1). To demonstrate the repeatability of the metahinge, we cyclically performed the activation and deactivation process. At intervals of every 10 cycles, we recorded the corresponding force–displacement relation for activation. This set of data is plotted as the experimental uncertainty domain shown in Fig. 1f. Plastic deformation may develop within the flexible hinges and stabilize after a certain number of cycles[32]. The 101st activation exhibits a limit force of 10.7 N, a value that is ~82% of that of the 1st activation. Despite this, the metahinge after cyclic usage can still promise a robust activated state, and the rotational ZM post-activation is well preserved.

## Isochoric reconfiguration from a honeycomb structure to a Kagome mechanism

The architecture shown in Fig. 1b can be treated as a one-dimensional bistable element that can be connected in a two-dimensional network to form a planar metamaterial capable of isochoric reconfiguration; this material is a unitary piece that can reversibly switch between a structure and a mechanism; its mechanisms of isochoric reconfiguration enable the arbitrary activation of selected metahinges without encountering geometry frustration, a distinct advantage over the existing literature[19,23,30]. To study this phenomenon, we first examine a network with nodal connectivity $Z = 3$, i.e., three bistable elements converging to each vertex (Fig. 2a). Each vertex of the network is a rigid joint, disallowing free rotation. In the initial state (see fabricated finite-period specimen at the bottom of Fig. 2a), the metamaterial resembles a hierarchical honeycomb structure with no ZMs. Upon a sequence of local activation forces (Supplementary Movie 2), the honeycomb structure undergoes an isochoric reconfiguration where all the metahinges are activated. We point out that the activation sequence has a negligible influence on the resulting activated state due to the isochoric characteristic of each metahinge. Therefore, any consecutive actions of activation are independent of each other. In the fully activated state, the honeycomb structure becomes a hinged Kagome mechanism featuring multiple floppy ZMs, as shown in Fig. 2b. In contrast to conventional multistable mechanical metamaterials[14,33], which undergo state transitions through boundary compression/tension, our metamaterial in its initial state (Fig. 2a) is highly robust in resistance to boundary loads, as a boundary action of compression/tension is unable to induce the required counter-rotation that can activate each pair of triangles. Its state transition is triggered only when local activation forces are applied to pairing hinges that are about to merge, a characteristic implying that its initial state is to some extent protected by local geometry and is insensitive to boundary perturbations.

To characterize how the activation of multiple metahinges influences the mechanical properties of the metamaterial, we performed

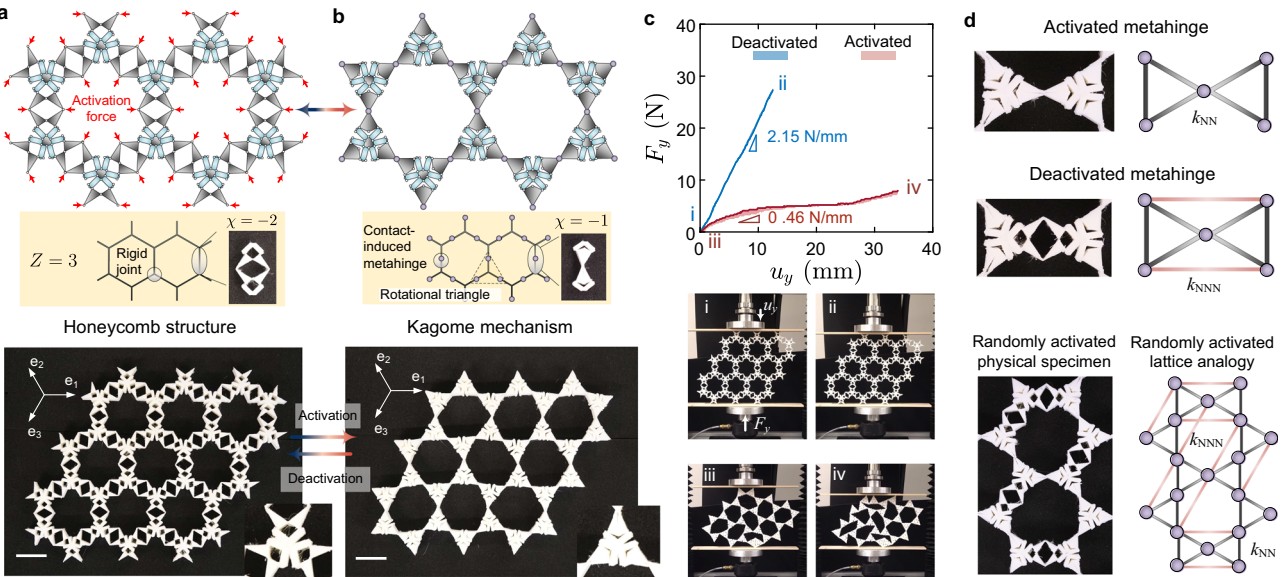

**Fig. 2 | Reprogramming zero modes in Kagome metamaterials through isochoric reconfiguration. a** Kagome metamaterial with all its metahinges being deactivated, resembling a honeycomb structure; white scale bar 44 mm; inset at the bottom-right corner illustrates how bistable elements are joined. **b** Fully activated Kagome metamaterial; white scale bar 44 mm; inset at the bottom-right corner is the fully activated rotation triangle. **c** Compression response of the metamaterial in its fully deactivated and activated state; experimental uncertain regimes were obtained by performing the compression test three times for the specimen in each of its states. **d** Lattice analogy and corresponding physical metamaterial.

compression tests on specimens in both their fully deactivated (initial) and activated states. Figure 2c illustrates their force–displacement curves with snapshots captured during the loading process. The fully deactivated metamaterial specimen behaves as a planar structure and undergoes a homogeneous bulk deformation mode (insets i and ii of Fig. 2c), i.e., an affine deformation response[34]; the effective compression stiffness in this deactivated state is 2.15 N/mm. Upon being fully activated, the metamaterial specimen becomes extremely floppy; it visibly wilts under self-weight when positioned on the compression platform (inset iii of Fig. 2c). Here the compression mainly leads to the localized deformation of the Kagome mechanism at the contact area (insets iii and iv of Fig. 2c), delivering an effective stiffness of 0.46 N/mm. In a nutshell, from the fully deactivated state to the fully activated state, the emergence of multiple ZMs enables a remarkable reduction in compression stiffness of ~79%, and the deformation changes from a homogeneous bulk mode (affine) to a localized (non-affine) mode[34] (Supplementary Movie 2).

## Lattice analogy for isochoric reconfiguration of Kagome metamaterials

To predict the number and pattern of ZMs in a Kagome metamaterial with arbitrarily activated metahinges, we now introduce a lattice analogy. The fully activated Kagome metamaterial (Fig. 2b) is a Maxwell lattice that can be represented by an assembly of hinged triangular frameworks[35], as shown at the top of Fig. 2d. Each vertex in the framework is a contact-induced metahinge allowing free (low-energy) rotation around its center. The black edges illustrated in Fig. 2d are the nearest neighbor (NN) bonds[35–37] connecting each metahinge to its nearest neighboring metahinge, showing axial stiffness $k_{NN}$. If the contact-induced metahinge is deactivated, the low-energy rotation vanishes, a scenario that is equivalent to adding a pair of the next nearest neighbor (NNN) bonds[35–39], with axial stiffness $k_{NNN}$, to the triangular frameworks (middle of Fig. 2d). The lattice analogy here presented serves to capture the number and pattern of ZMs as opposed to accurately predicting the actual elasticity and inertial characteristics of the metamaterial; for this reason the value of both

$k_{NN}$ and $k_{NNN}$ is now set to unity. Fig. 2d shows the relationship between a randomly activated physical specimen and the corresponding lattice analogy, which can be leveraged to determine the number ($N_0$) and pattern of ZMs through the calculation of the null space of the kinematic matrix (Supplementary Note 2).

## Uniaxial and biaxial zero modes through selective activation of metahinges

A fully activated Kagome metamaterial has multiple infinitesimal ZMs along its network axes, characterized by a collective counter-rotation between pairs of triangular frameworks[35]. Due to this strong kinematic indeterminacy, multiple deformation pathways[40] are possible in a fully activated Kagome metamaterial (Fig. 2b). To deterministically enable one deformation pathway to emerge, we need to selectively incorporate NNN bonds so as to degenerate the Kagome metamaterial to a single-degree-of-freedom (SDOF) system, i.e., a lattice framework possessing only one ZM. Illustrated in Fig. 3a is a representative lattice analogy with only one ZM traveling along the $e_1$ axis, whose displacement vector is denoted as $\Phi_{e_1}$ and depicted by the red arrows. The physical counterpart of the lattice is shown on the right of Fig. 3a, where a rotational input on the right can be successfully delivered to the opposite end (left) through the designed ZM. We emphasize that this uniaxial ZM is infinitesimal; hence under a finite deformation, the rotation signal exhibits a decay from the source (right) to the output end (left). This type of uniaxial ZMs for mechanical signal transmission can also be realized individually along the $e_2$ and $e_3$ axes.

We now showcase how to construct a biaxial ZM by leveraging two individual uniaxial ZMs. We first discuss the linear superposition of two ZMs along the $e_i$ and $e_j$ ($i \neq j$) axes. It has been recently demonstrated that in a rotation-based metamaterial, a rotation node, where the relative rotation between two adjacent bodies vanishes, can arise due to the superposition of two ZMs at their intersection boundary[41]. A similar phenomenon can be captured in the Kagome lattice system shown in Fig. 3b. Here NNN bonds have been strategically added to the system which has now acquired two uniaxial ZMs running through the

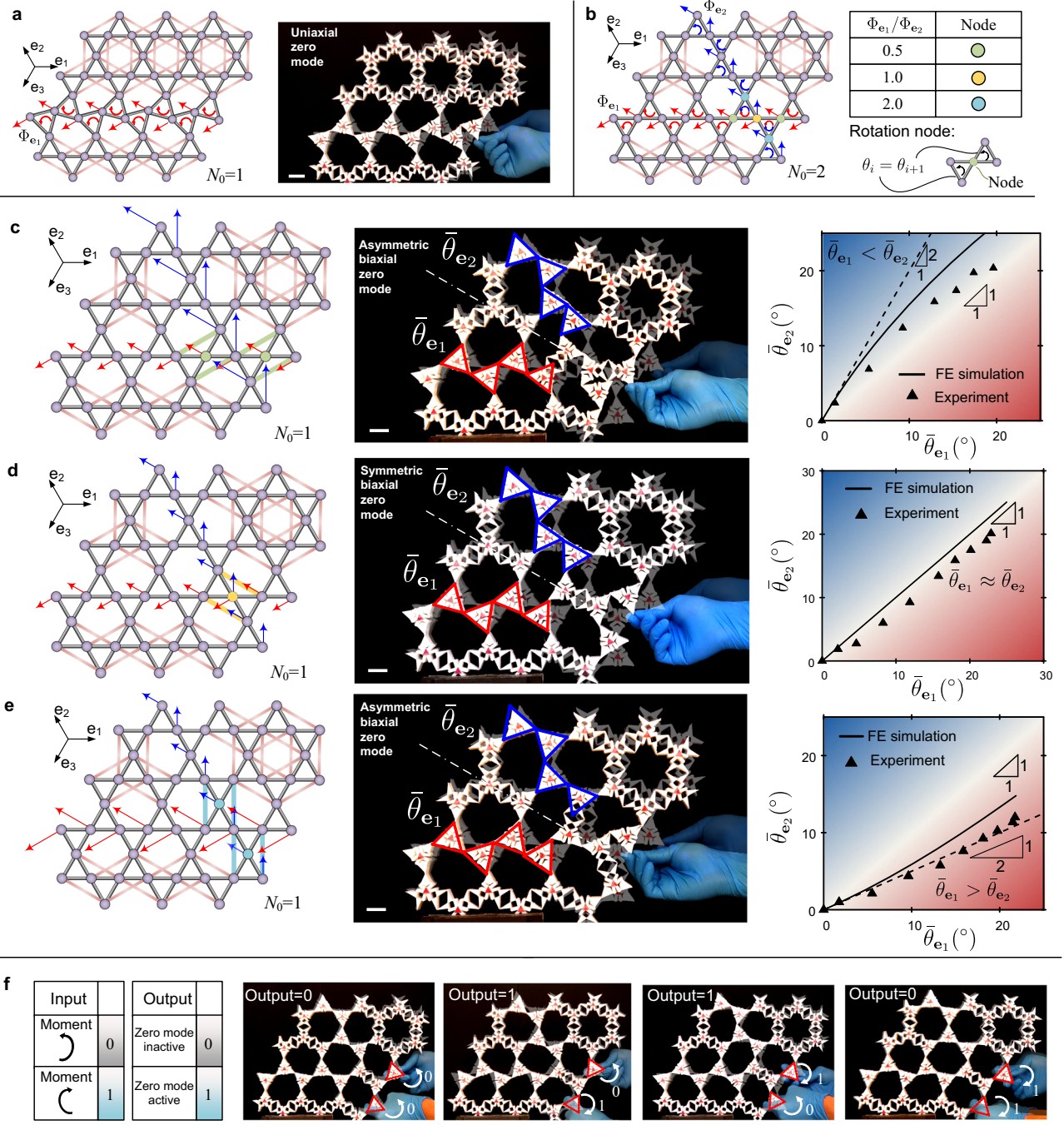

**Fig. 3 | Uniaxial and biaxial zero modes in a selectively activated Kagome metamaterial. a** Selectively activated Kagome metamaterial showing a uniaxial zero mode; $N_0$ is the number of zero modes of the lattice analogy; white scale bar 22 mm. **b** Linear superposition of two uniaxial zero modes and possible rotation nodes. **c**–**e** Biaxial zero modes featuring symmetry and asymmetric deformation with respect to the angle bisector in white; from top to bottom, selectively activated specimens can deliver $\bar{\theta}_{\mathbf{e}_1} < \bar{\theta}_{\mathbf{e}_2}$, $\bar{\theta}_{\mathbf{e}_1} \approx \bar{\theta}_{\mathbf{e}_2}$, and $\bar{\theta}_{\mathbf{e}_1} > \bar{\theta}_{\mathbf{e}_2}$ respectively; white scale bar 22 mm. **f** Mechanical XOR gate blocking the propagation of in-phase rotational signals and transmitting out-of-phase signals.

$\mathbf{e}_1$ and $\mathbf{e}_2$ axes, denoted by the normalized displacement vectors $\Phi_{\mathbf{e}_1}$ and $\Phi_{\mathbf{e}_2}$ respectively. A linear combination of $\Phi_{\mathbf{e}_1}$ and $\Phi_{\mathbf{e}_2}$, given by $c_1\Phi_{\mathbf{e}_1} + c_2\Phi_{\mathbf{e}_2}$, constitutes another ZM. Depending on the value of $c_1/c_2$, there exist three types of superpositioned ZMs featuring three types of rotation nodes, marked in Fig. 3b with distinct colors. If $c_1/c_2 = 0.5$, for example, the superpositioned ZM features two rotation nodes highlighted in green; the adjacent triangles around each green metahinge exhibit an identical rigid-body motion, a phenomenon implying that this green metahinge is inactive. As a result, adding two pairs of NNN bonds around these two inactive metahinges has no influence on this superpositioned ZM (left of Fig. 3c where the added NNN bonds are colored in green as the corresponding rotation node). Adding such NNN bonds can kill the original two ZMs, $\Phi_{\mathbf{e}_1}$ and $\Phi_{\mathbf{e}_2}$, while preserving the superpositioned ZM, $0.5\Phi_{\mathbf{e}_1} + \Phi_{\mathbf{e}_2}$, a phenomenon that characterizes the unique ZM of the lattice system, i.e., the lattice framework degenerates to a SDOF system. This functionality can be leveraged to create a mechanical signal transmitter that can couple motions along two distinct axes, as described below.

The experimental results in Fig. 3c, d, and e (see also Supplementary Movie 3) attest the attainment of dissimilar modal amplitudes along two axes. For example, to demonstrate the biaxial ZM in Fig. 3c, where the rotation amplitude is larger along the $\mathbf{e}_2$ axis than the $\mathbf{e}_1$ axis, we first extract the four triangles outlined in red; then we evaluate the average of their absolute rotation to represent the modal amplitude along the $\mathbf{e}_1$ axis, which is denoted as $\bar{\theta}_{\mathbf{e}_1}$. Similarly, the four triangles outlined in blue in Fig. 3c (middle) are used to evaluate $\bar{\theta}_{\mathbf{e}_2}$, the modal amplitude along the $\mathbf{e}_2$ axis. On the right of Fig. 3c, we compare the experimentally obtained $\bar{\theta}_{\mathbf{e}_1} - \bar{\theta}_{\mathbf{e}_2}$ relationship with that obtained from nonlinear Finite Element (FE) simulations of the lattice analogy (Supplementary Note 3). The black dashed line, with a slope of 2.0, represents the $\bar{\theta}_{\mathbf{e}_1} - \bar{\theta}_{\mathbf{e}_2}$ relation under the assumption of infinitesimal deformation. The experimentally obtained data closely match the FE simulation results in the small-deformation regime while exhibiting a deviation as the deformation increases. This slight overestimation of $\bar{\theta}_{\mathbf{e}_1}$ obtained from the FE simulation is mainly attributed to the assumed value (unity) of the axial stiffness of the NN and NNN bonds in the lattice analog model. Despite this discrepancy, the experimental results confirm that for this biaxial ZM, we can obtain $\bar{\theta}_{\mathbf{e}_1} < \bar{\theta}_{\mathbf{e}_2}$. On the other hand, by introducing the corresponding NNN bonds around the yellow or blue rotation nodes (left of Figs. 3d and 3e respectively), we can acquire other two types of biaxial ZMs. One (Fig. 3d) exhibits a nearly symmetric deformation with respect to the angle bisector of two axes (white dashed line), i.e., $\bar{\theta}_{\mathbf{e}_1} \approx \bar{\theta}_{\mathbf{e}_2}$, whereas the biaxial ZM in Fig. 3e has $\bar{\theta}_{\mathbf{e}_1} > \bar{\theta}_{\mathbf{e}_2}$.

The results obtained above can now be leveraged for mechanical logic operations. A demonstrative example is the realization of an XOR mechanical logic gate using the biaxial ZM shown in Fig. 3d. The two independent inputs of the logic gate are the moments applied to the two rotational triangles outlined in red; their values are interpreted on the left of Fig. 3f. The output of the logic gate is contingent on the status of the biaxial ZM. If the ZM is active, allowing the mechanical signal to be transmitted, we define the output as 1; in contrast, if the ZM is inactive, the mechanical signal is blocked, implying an output of 0. As illustrated in Fig. 3f, only if the two input moments are out-of-phase, i.e., spinning conversely, the ZM can be activated, yielding an output of 1. Otherwise, the input moments can merely lead to an incompatible deformation localized at the area where the moments are applied. By virtue of this logical characteristic, selected metahinges can be activated in a metamaterial to act as a mechanical transmission system capable of filtering unwanted signals depending on whether the input signals are in-phase or out-of-phase (Supplementary Movie 4).

## All-in-one reprogrammable architecture transforming into either a structure, a multistable matter, or a mechanism

A fully deactivated Kagome metamaterial represents a stable structure, whereas a fully activated Kagome metamaterial is an unstable mechanism. A natural question arises: can a Kagome metamaterial with a selectively activated portion of its metahinges become metastable, i.e., is it possible to selectively switch the local state of each bistable metahinge to activate the global multistability of the metamaterial? To address this matter, we first analyze the similarity between a Kagome lattice with selectively added NNN bonds (Fig. 4a) and the fully deployed state of a planar kirigami with triangular motifs[42]. In the lattice analogy (Fig. 4a), the red NNN bonds and their associated triangles act akin to the Y-shaped elastic confinement observed in an existing planar kirigami[42]; the remaining triangular sub-frameworks in the lattice analogy resemble the rotational bodies in that planar kirigami. It has been approved that the unit cell of the planar kirigami has two stable states, one deployed and one collapsed[42]. Given the direct geometric similarity, we can anticipate that the lattice analogy presented in Fig. 4a can also deliver an auxetic bistable transition from the deployed state to a collapsed state. To verify the bistability of this transition, we employ the Nudged Elastic Band (NEB) method[43,44] and probe the minimum energy path (MEP) during state transition. If an unavoidable energy barrier exists along the MEP, the transition is bistable[45]. Otherwise, this transition corresponds to a finite-amplitude ZM or a finite collapse mechanism[5]. In the NEB method, we initialize the state transition path with a linear interpolation between the deployed and collapsed states, serving as our initial guess for the MEP. We iteratively update the state transition path (Supplementary Note 4) and examine how the energy landscape evolves, as plotted in Fig. 4b, where $A$ and $\Delta A$ stand for the initial area and area change of the unit cell respectively. We observe that the energy landscape eventually converges to the MEP represented by a red curve with a non-zero energy barrier, attesting that this state transition of the lattice analogy is bistable. Fig. 4c and d illustrate two key states of the periodic lattice analogy: state ii, the local maxima of the MEP, and state iii, the fully collapsed state. The normalized area changes, $\Delta A/A$, in states ii and iii are $-0.41$ and $-0.75$, respectively.

To experimentally demonstrate the bistable/multistable transition in a physical specimen selectively activated in the manner mentioned above, we extract a metastrip of the periodic lattice analogy, as indicated by the purple region in Fig. 4a. A fully deactivated metastrip is a structure with no ZMs (Fig. 4e). A selectively activated metastrip resembles the deployed state of the planar kirigami with triangular motifs, as shown on the left of Fig. 4f. By sequentially overcoming the geometry incompatibility/energy barrier of the unit cells (Supplementary Movie 5), we can transition the metastrip to a stable collapsed state (right of Fig. 4f) with an area change $\Delta A/A = -0.65$; this value is in between $-0.41$ and $-0.75$ since the metastrip has exceeded the local maxima (Fig. 4c) on the energy landscape but is prevented from reaching the fully collapsed state (Fig. 4d) due to the presence of internal contact, or in another word, due to the intersection between NN and NNN bonds in the lattice analogy. This experimental result demonstrates that the metamaterial can undergo reversible transformations that embody its architecture with the traits of either a structure (Fig. 4e), a multistable matter (Fig. 4f), or a mechanism (Fig. 4g).

## Reprogrammable phonon spectra

Besides studying the ZMs of a selectively activated Kagome metamaterial in the static regime, we now employ the lattice analogy to demonstrate the reprogrammability of its linear phonon spectra. Illustrated at the top of Fig. 4h–j are three representative periodic lattices, each tessellated through their unit cell (highlighted in yellow) along the lattice vectors $\mathbf{a}_1$ and $\mathbf{a}_2$. The unit cell has selectively added NNN bonds that allow ZMs to propagate along designated axes (red arrows). By applying Bloch's theorem and varying the wave vector within the first Brillouin zone, we can solve the eigenfrequency of this harmonic system and obtain its linear phonon spectra (Supplementary Note 5).

Figure 4 h–j illustrates the first five normalized eigenfrequencies, denoted by $\omega_i^* (i = 1, \ldots, 5)$, and corresponding density plots of each period lattice. The existence of zero frequencies is dependent on the wave vector $[k_x, k_y]$[38]. For example, the lattice illustrated in Fig. 4a, a periodic counterpart of the specimen in Fig. 3a, has a zero-frequency contour defined by $k_x = 0$ (reciprocal space) within the lowest frequency branch $\omega_1^*$; the eigenvectors on this contour manifest localized deformation modes along the $\mathbf{e}_1$ axis in direct space, and the phase velocity along this contour is zero. In total, three acoustic branches originate from $[k_x, k_y] = [0, 0]$; two of them correspond to the longitudinal and shearing bulk waves, while the remaining one represents a localized wave along the $\mathbf{e}_1$ axis. Once we remove relevant NNN bonds to activate another ZM along the $\mathbf{e}_3$ axis (Fig. 4i), there will be two zero-frequency contours characterized by $k_x = 0$ and $k_x = -\sqrt{3}k_y$ respectively. Now the periodic lattice manifests four acoustic branches starting at $[k_x, k_y] = [0, 0]$; this lattice exhibits strong anisotropy around

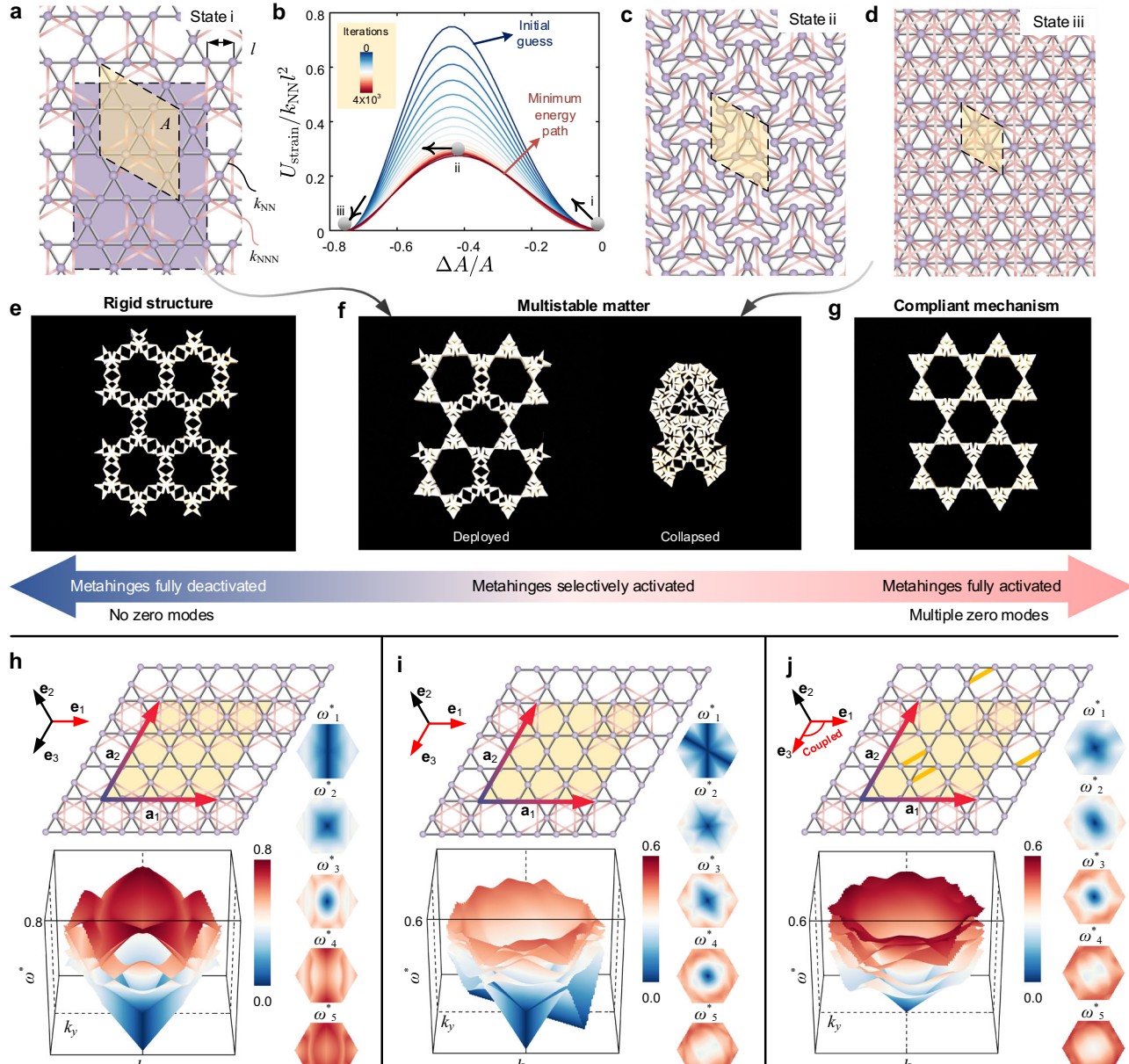

**Fig. 4 | Selective activation of metahinges for reversible transformation among a structure, a multistable matter, and a mechanism (top); reprogrammable phonon spectra (bottom). a** Periodic Kagome lattice with selectively added the next nearest neighbor (NNN) bonds resembling the fully deployed state of planar multistable kirigami sheets[42]. **b** Iterative optimization of energy landscape during state transition leveraging the Nudged Elastic Band method, confirming that the periodic lattice is multistable. **c** Configuration of the periodic lattice at the energy local maxima. **d** Fully collapsed state of the lattice analogy. **e–g** Physical metastrip specimen capable of transforming its architecture into a structure (**e**), a multistable matter (**f**), and a mechanism (**g**). **h–j** Phonon spectra of periodic lattices with selectively added NNN bonds; $\omega^*$ is the normalized eigenfrequency; $\mathbf{e}_i$ axes in red indicate activated directions allowing zero modes to propagate; the bottom-left corner of each plot is the band diagram; the right column is the density plots of each eigenfrequency branch.

the $[k_x, k_y] = [0, 0]$ point (Supplementary Note 5). Upon adding a pair of NNN bonds (yellow bonds in Fig. 4j) around the intersection node of two axes, we can eliminate the original two zero-frequency contours along $k_x = 0$ and $k_x = -\sqrt{3}k_y$, and create a new acoustic branch manifesting a coupled deformation mode along the $\mathbf{e}_1$ and $\mathbf{e}_3$ axes in direct space. This emerging acoustic branch exhibits a non-zero phase velocity at $[k_x, k_y] = [0, 0]$, with the minimum phase velocity occurring at the angle bisector of $k_x = 0$ and $k_x = -\sqrt{3}k_y$. Additionally, the phase velocity around the $[k_x, k_y] = [0, 0]$ point is less dependent on the direction of the wave vector, suggesting that the selective addition of NNN bonds can be employed to reprogram the anisotropy of the lattice in the low-frequency regime.

## Reprogramming buckling modes in a rotation-square metamaterial via selective activation of metahinges
As well known, the buckling behavior of a planar structure is governed by the competition between the in-plane stretching/compression energy and the out-of-plane bending energy[46]. Given our metamaterial concept allows for reprogrammable in-plane ZMs capable of modifying the ratio between the in-plane and out-of-plane energy, we anticipate that the selective activation of metahinges might offer a means to suppress the out-of-plane buckling of the planar metamaterial.

As a demonstrative example, we form a square network with $Z = 4$ by connecting our bistable element (Fig. 1b). As illustrated in Fig. 5a, the network in its initial state is a rigid structure ($N_0 = 0$). As the

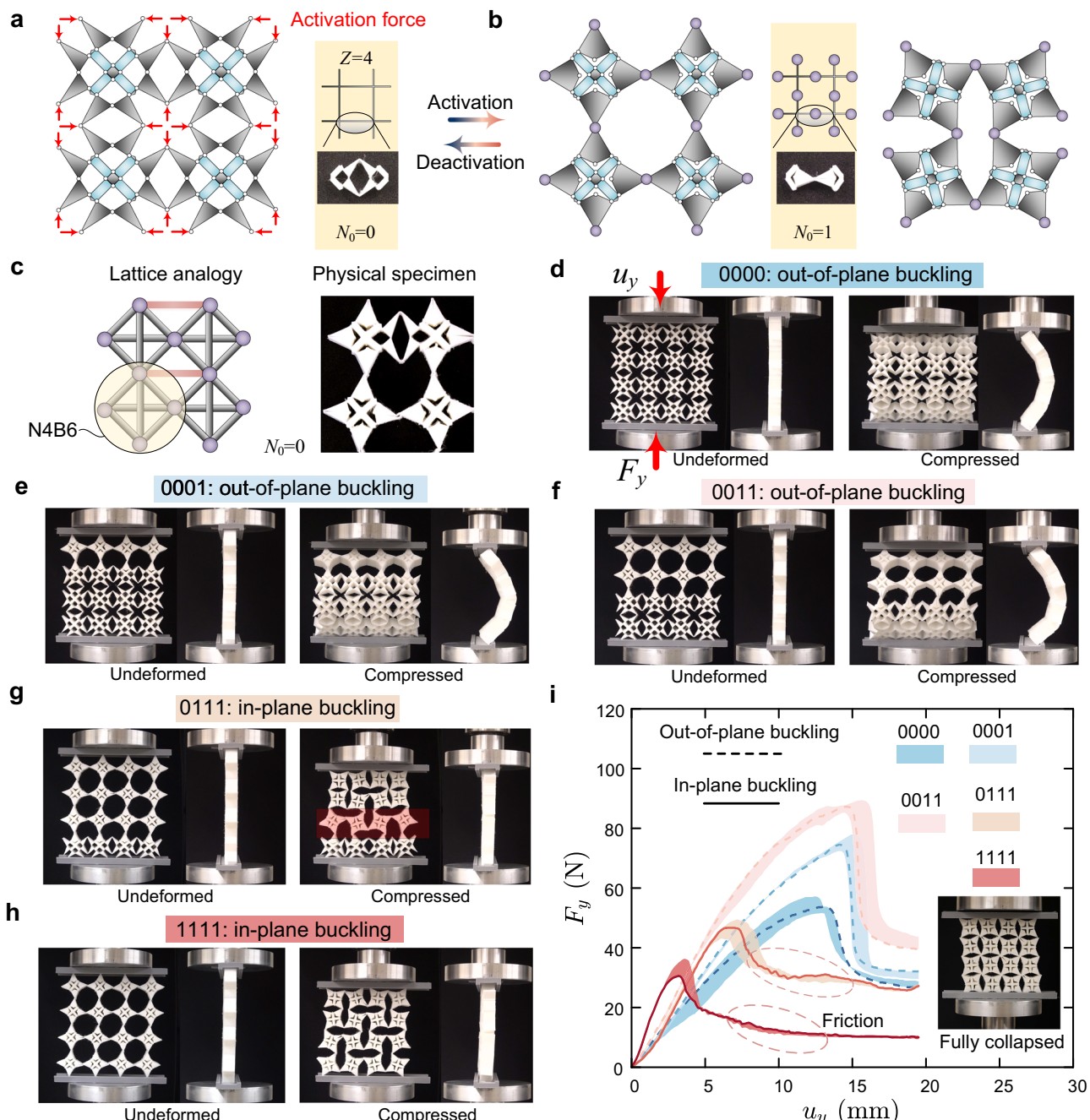

**Fig. 5 | Reprogramming buckling modes in a rotation-square metamaterial.**
**a** Square network in its fully deactivated state as a rigid structure. **b** Fully activated square network featuring one bulk zero mode. **c** Lattice analogy representing a selectively activated metamaterial; the activated rotation square can be described by four nodes connected via six bonds (N4B6 model). **d–h** Snapshots of compressed metamaterial specimens in five distinct activated states; in each plot, the left and right columns display the undeformed and deformed configurations

respectively; both front and side views are included; each state of the specimen is labeled with a four-digit binary number, with each digit denoting whether the corresponding row is activated (1) or deactivated (0). **i** Force–displacement relations of selectively activated rotation-square metamaterials; dashed curves correspond to states exhibiting out-of-plane buckling, while solid curves stand for states showing in-plane buckling; friction-induced noise can be witnessed in states 0111 and 1111 due to the sliding between rotation squares and compression indenters.

contact-induced metahinges are activated in the middle of each bar, the initially kinematically determinate structure turns into a rotation-square mechanism[47], giving rise to an auxetic bulk ZM[48]. As shown on the right of Fig. 5b, this bulk ZM is a finite-amplitude mechanism persisting in a large-deformation regime until the gaps between the rotation squares are fully closed. The rotation-square mechanism can be represented by a lattice analogy, where the fully activated rotation square is represented by a framework comprising four nodes and six bonds (Fig. 5c), namely the N4B6 model[49]. Similarly, the deactivation of the contact-induced metahinge is equivalent to adding a pair of bonds

(highlighted in light red) to eliminate the relative free rotation between two rotation squares. It is evident that the lattice analogy becomes an SDOF mechanism only if there are no red bonds, i.e., the metamaterial is fully activated. Therefore, a partially activated metamaterial exhibits a response falling between a thick plate structure (fully deactivated) and an SDOF mechanism (fully activated), where the former buckles out-of-plane upon reaching the critical point, and the latter maintains in-plane deformation even under significant compression.

Figure 5 d–h presents snapshots of a compressed metamaterial specimen across five distinct activated states, and Fig. 5i shows their

corresponding force–displacement relations. In the fully deactivated state 0000, the planar structure initially exhibits a monotonic force–displacement response; as the displacement reaches ~13.3 mm, the specimen suddenly buckles out-of-plane and undergoes a snap-back instability, a limit-point buckling phenomenon generally observed in thick plates or wide beams[50]. In the partially activated states 0001 and 0011, the internal contact strengthens the effective bending stiffness of the local Timoshenko-type beams[19], a phenomenon that improves to a certain extent the global compressive stiffness of the specimen prior to the onset of instability. The out-of-plane buckling, however, still exists, and the corresponding critical displacement has rarely changed. Up to this point, the out-of-plane bending is still the energy-favored deformation mode in the post-buckling regime.

An opposite outcome to the above is obtained if the top three rows are activated (Fig. 5g). The out-of-plane buckling is now suppressed, but the local buckling of metahinges[51] formed by self-contact is triggered at ~6.7 mm of compression, which leads to a discontinuous in-plane post-buckling response[52]. Comparing Fig. 5d–g, we observe that as more rotation squares are activated, the buckling mode transitions from the global out-of-plane bending of the entire specimen to the in-plane bending of the contact-induced metahinges: the latter replaces the former as the energy-favored mode in the post-buckling regime. The critical displacement exhibits a substantial decrease as the buckling mode switches from out-of-plane to in-plane, primarily due to the characteristic dimension of the contact-induced metahinge which is much smaller than that of the entire specimen. Since the bulk ZM is not fully activated in the 0111 state, the auxetic phenomenon does not propagate across the entire specimen. As a result, a domain wall (shaded in red) emerges on the third row, delineating a boundary between the auxetic region (top) and the non-auxetic region (bottom), namely the mechanism region (top) and the structure region (bottom). This type of domain wall has been found in rotation-square metamaterials with intentionally introduced pinning defects[53]. In our metamaterial, the deactivation of a contact-induced metahinge is analogous to introducing a pinning defect. Our approach to introducing defects, however, is rooted in the bistability of the local architecture, eliminating the need for manual adhesion to immobilize the defect[53]. As a result, the versatility of our strategy enables a more convenient reprogrammability of the domain wall within a rotation-square metamaterial.

Finally, if all the metahinges have been activated, the critical displacement and force exhibit a further reduction (red curve in Fig. 5i). This is due to the complete activation of the auxetic ZM, making the entire system less stable yet with no out-of-plane buckling. The specimen now transforms into a compliant rotation-square mechanism, and its fully collapsed state is shown in the inset of Fig. 5i. In summary, this set of results shows that through the progressive activation of the rotation-square metamaterial, we can on-demand suppress the out-of-plane buckling and trigger the in-plane buckling of metahinges at a smaller scale, thus concurrently reprogramming the associated critical forces and displacements (Supplementary Movie 6).

### Generalization to metamaterials comprising metahinges with a higher coordination number

In the context of a fully activated metamaterial, the coordination number is defined as the average number of rotation bodies connected at each activated metahinge, e.g., the fully activated metamaterials in Figs. 2b and 5h have a coordination number of two. Here, we demonstrate the creation of a metahinge featuring a higher coordination number, enabling the emergence of a larger number of ZMs upon activation. The strategy here pursued is to admit higher-order cyclic symmetry in the metahinge architecture so as to access a larger pool of metamaterial tessellations with higher coordination numbers.

The metahinge architecture illustrated in Fig. 1b exhibits $C_2$ cyclic symmetry yielding a corresponding set of metamaterial tessellations. If

we extract its left half to create a new assembly featuring $C_3$ symmetry, we can obtain an activated metahinge with a coordination number of three, as shown in Fig. 6a. This architecture can then be connected to form a Kagome-type metamaterial belonging to the *p6 mm* group[54] (Fig. 6b). The activation process in this case is non-isochoric. The area of the activated unit cell shrinks to about 53% of its original value (shaded yellow regions in Fig. 6b). The metamaterial is also able to deliver a dual response: the effective compressive stiffness, which is 4.96 N/mm in the fully deactivated state, decreases to 0.46 N/mm due to the emergence of numerous ZMs post-activation (Fig. 6c and Supplementary Movie 7). The fully activated specimen resembles a hinged honeycomb (inset iii of Fig. 6c) and hence is highly flexible, leading to an initial deformation under gravity, similar to the Kagome metamaterial shown in Fig. 2c. In this case, internal contact between pairing rotation bodies[27] emerges in the specimen lower part due to the small spacing angle resulting from the high coordination number. This densification in the lower part results in a concentrated deformation at the upper edge of the specimen (insets iii and iv in Fig. 6c).

The type of non-isochoric transformation described above can also be interpreted through our lattice analogy. The fully deactivated state of the metamaterial (left of Fig. 6b) can be considered as a network hinged by a series of constituent rectangles characterized by an aspect ratio of $s_1/s_2$. A typical constituent rectangle (shaded in purple in Fig. 6b) can be represented by the N4B6 model in the lattice analogy (left of Fig. 6d). For non-isochoric transformations, activating the metahinges is equivalent to reducing the aspect ratio $s_1/s_2$ until it approaches zero. As $s_1/s_2 \to 0$, the rectangle degenerates to a bond, and the lattice analogy transforms into a hinged hexagonal lattice (Fig. 6d). This non-isochoric transformation leads to a significant change in the phonon spectrum: the acoustic branch with the lowest group velocity degenerates to a zero-frequency contour, the bandgap (gray area) between the fourth and fifth branches is broadened, and a bandgap (blue area) between the ninth and tenth branches emerges (Supplementary Note 6). Overall, the band structure shifts to a lower frequency regime due to the softening behavior endowed by the metahinge activation.

Similarly, we can explore higher-order of cyclic symmetry, and create for example an architecture with $C_4$ symmetry, featuring a coordination number of four in the fully activated state, as illustrated in Fig. 6e. The architecture can then be tessellated to build a square-type metamaterial (left of Fig. 6f). In the fully deactivated state, the metamaterial specimen already exhibits one bulk auxetic ZM (insets i and ii Fig. 6g), and hence is comparably floppy under compression (blue curve in Fig. 6g). Upon full activation, the metamaterial specimen manifests an increased number of ZMs and loses its shearing resistance. Affected by gravity and imperfections, the specimen is tilted to one side, resulting in a sequence of internal contacts between rotation bodies (insets iii and iv in Fig. 6g); this internal contact further alters the specimen topology[27] and substantially improves the compression stiffness (red curve in Fig. 6g and Supplementary Movie 8). Fig. 6h shows the lattice analogy for this *p4mm* group metamaterial. The original rectangle-hinged lattice degenerates to a square lattice upon activation. Zero frequencies emerge along the Γ–Y contour in reciprocal space, corresponding to the shearing ZMs in direct space. Similarly, the activation of metahinges can lead to the emergence and vanishing of bandgaps (Supplementary Note 6), as shown in Fig. 6h. We remark that the spacing angle $\tau$ in the fully activated state (Fig. 6a, e) is a crucial geometry parameter preventing us from reaching higher-order cyclic symmetry; $\tau$ decays exponentially with the symmetric order, and hence the internal contact between rotational bodies would disrupt the bistable activation process. While the metamaterial tessellations incorporating $C_3$ and $C_4$ metahinges (Fig. 6) enable a reprogrammable number of ZMs, facilitating in situ stiffness tuning, their global response lacks the ability to transition between being multistable and monostable, unlike the Kagome metamaterial depicted in Fig. 4e–g.

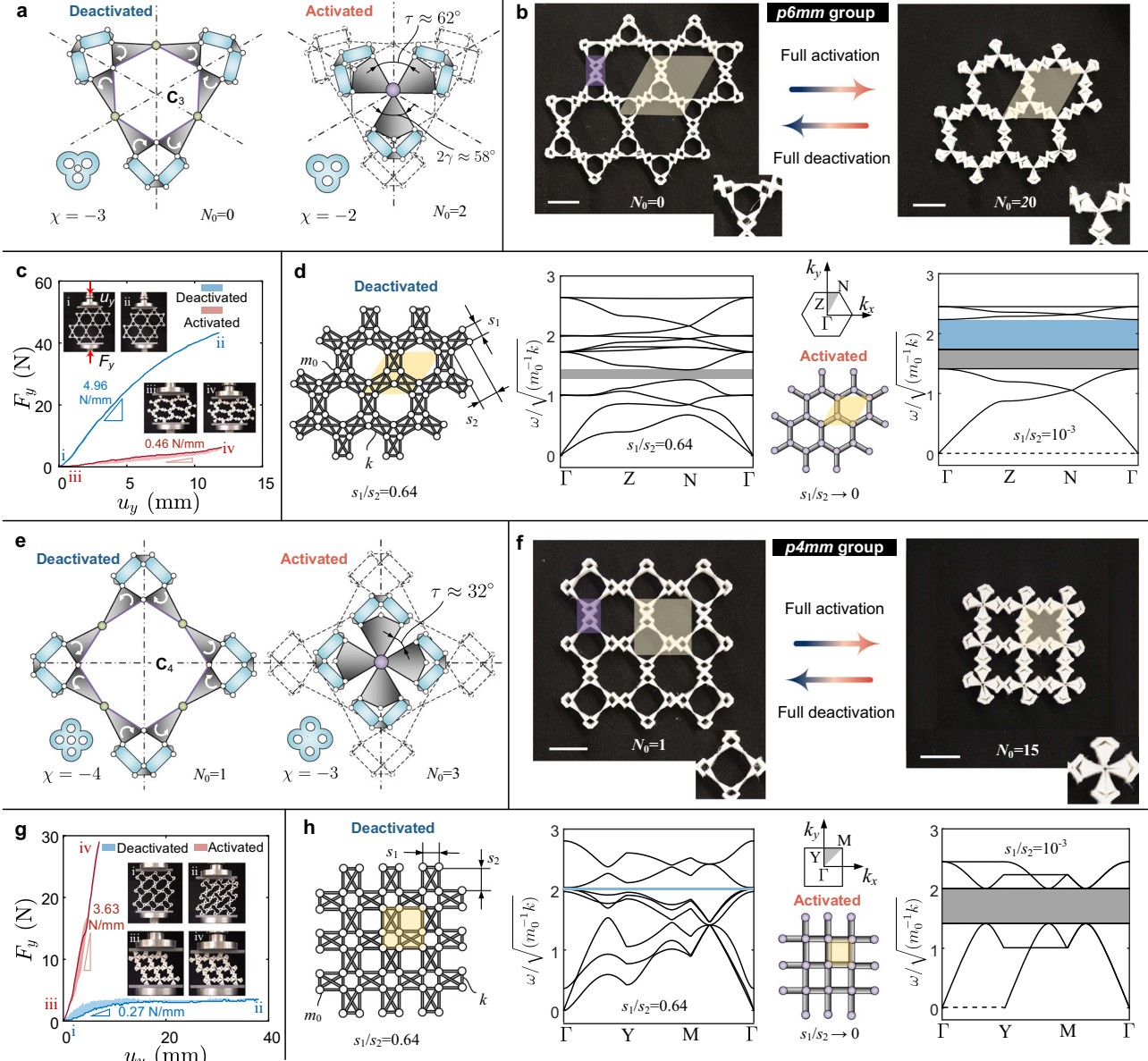

**Fig. 6 | Metamaterials comprising metahinges with a higher coordination number. a** Activation of the metahinge with $\mathbf{C}_3$ symmetry showing a coordination number of three; the Euler's characteristic $\chi$ transitions from −3 to −2. **b** Non-isochoric reconfiguration from a Kagome-type metamaterial to a hinged honeycomb; region shaded in yellow is the unit cell; region shaded in purple is the constituent rectangle that is kinematically determinate; insets at the bottom-right corner illustrate the topological transformation of the metahinge; white scale bar 44 mm. **c** Compression responses of fully deactivated and activated *p6mm* metamaterial specimens. **d** Lattice analogy and phonon spectra for the *p6mm*

metamaterial; the first Brillouin Zone is shown at the top of the third column; the dashed line is the zero-frequency contour. **e** Activated metahinge with $\mathbf{C}_4$ symmetry showing a coordination number of four; the Euler's characteristic $\chi$ transitions from −4 to −3. **f** Non-isochoric reconfiguration from a rectangle-hinged metamaterial to a square lattice; white scale bar 44 mm. **g** Compression responses of corresponding fully deactivated and activated *p4mm* metamaterial specimens. **h** Lattice analogy and phonon spectra for the *p4mm* metamaterial; the original bandgap shaded in blue vanishes upon full activation.

## Discussion

In summary, we have presented a class of mechanical metamaterials that feature reprogrammable zero modes enabled by the selective activation of metahinges. The architecture of the metahinge is rationally redefined over the existing literature to tackle the challenge of geometry frustration that arises during its progressive activation in a tessellated two-dimensional metamaterial. This allows our Kagome-type metamaterial to reversibly transition among a rigid structure, a compliant mechanism, and a multistable matter, hence integrating their conflicting mechanical characteristics within a single topology-transformable architecture. We also showcase the generalization of our concept to a rotation-square metamaterial and

the creation of a metahinge with higher cyclic symmetry. In distinct activated states, this class of metamaterials is able to deliver stiffness reprogramming near one order of magnitude, adjustable phonon spectra, on-demand buckling mode suppression, and switchable auxeticity, providing a promising avenue for the development of all-in-one devices for application in a diverse range of engineering fields. Finally, we envision the scalability of our metahinge within a range spanning from millimeters to meters and the possibility of embedding the metahinge architecture into origami-type metamaterials, enabling on-demand activation of foldability, thus extending our concept from two-dimensional planar materials to three-dimensional bulk materials.

## Methods

### Fabrication

We fabricated all the experimental specimens through Fused Deposition Modeling (Anycubic Vyper, China). The constituent material is thermoplastic polyurethane (TPU) 95A featuring a large elongation at break, ensuring that the printed flexible hinges can repeatedly undergo finite rotation without damage. The in-plane dimensions of the specimen presented in Fig. 1f can be found in Supplementary Notes 1 and 7; its out-of-plane thickness is 30 mm. The finite-period specimens illustrated in Fig. 2 and 6 were assembled by multiple printed pieces via interlocking mechanism and adhesion (see Supplementary Note 7 for more details); their out-of-plane thickness is 25 mm. The rotation-square metamaterial shown in Fig. 5 was printed integrally without any assembly. All the specimens were printed at their fully deactivated configuration. To ensure the accuracy and efficiency of the printing process, we set the layer height, line width, infill density, and printing speed as 0.20 mm, 0.26 mm, 100%, and 80.00 mm/s, respectively.

### Compression test

The quasi-static compression tests illustrated in Figs. 1f, 2c, 5i, 6c, and 6g were performed on Bose ElectroForce 3510 (Bose Corporation, Framingham, Massachusetts). Given the distinct compression ratio each specimen can undergo, we adjusted the loading rate accordingly such that the quasi-static condition is satisfied, and the total loading time for each specimen is controlled within a reasonable range. For example, the metamaterial samples bearing bulk zero modes (the fully activated Kagome metamaterial shown in Fig. 2b and the deactivated metamaterial in Fig. 6f) can undergo large compression, and hence the loading rate is moderately increased to reduce the total running time. The activation of the metahinge was at a loading rate of 0.68 mm/s (Fig. 1f). The compression for the fully deactivated and activated Kagome metamaterial (Fig. 2c) was at a loading rate of 0.52 mm/s and 1.92 mm/s, respectively. The rotation-square metamaterial was compressed at a rate of 1.40 mm/s (Fig. 5i). The metamaterial bearing $C_3$ metahinges was compressed at a rate of 0.52 mm/s in both its deactivated and activated states (Fig. 6c). The metamaterial bearing $C_4$ metahinges in its deactivated and activated states was compressed at a rate of 1.92 mm/s and 0.30 mm/s, respectively. Each specimen except for the metahinge in each of its activated states, was tested three times to obtain the experimental uncertainty regime, illustrated as the shaded areas in the plots; we selected one representative result to be plotted as a solid curve.

### Mechanical signal guiding through biaxial zero modes

Equilateral triangular markers in red are attached to the center of each rotation body to trace the local rotation of the biaxial zero modes in the selectively activated Kagome metamaterial. The bottom-left corner of the finite-period specimen is clamped to freeze global rigid-body motions. A slow push is applied to the free edge of the rightmost rotation body to trigger the biaxial zero mode. The quasi-static deformation process is recorded by Sony RX 100 (Sony Corporation, Japan) with a sampling frequency of 60 Hz. Representative frames are extracted from the video at intervals of 1.67 s and imported into Microsoft Visio (Microsoft Corporation, United States). Masks are added to the red triangular markers of interest (see the outlined triangles in Fig. 3c–e). The modal amplitudes along the $\mathbf{e}_1$ and $\mathbf{e}_2$ axes are then evaluated as the average rotation of the corresponding masks, denoted by $\bar{\theta}_{\mathbf{e}_1}$ and $\bar{\theta}_{\mathbf{e}_2}$ respectively.

## Data availability

All the data supporting the conclusions of this study are included in the article and the Supplementary Information file. Source data are provided with this paper.

## Code availability

The codes that support the findings of this study are available from the corresponding author upon request.

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

## Acknowledgements

D.P. acknowledges financial support from the Canada Research Chairs Program and the Natural Sciences and Engineering Research Council of Canada. L.W. acknowledges financial support from the China Scholarship Council (202006280037).

## Author contributions

L.W. and D.P. conceived the research and metamaterial design. L.W. carried out the analyses and experiments under the close direction and supervision of D.P., L.W. and D.P. analyzed and interpreted the results as well as wrote and revised the manuscript.

## Competing interests

The authors declare no competing interests.
