## [Peer Review File · Nature Communications]

Zero modes activation to reconcile floppiness, rigidity, and multistability into an all-in-one class of reprogrammable metamaterialsREVIEWER COMMENTS

Reviewer #1 (Remarks to the Author):

This excellent manuscript includes the investigation of a metamaterial design that exploits the use of multi-stability to reversibly switch between a floppy mechanism and a stiff structure. A comprehensive experimental study is undertaken that uncovers how the concept can be employed to switch between multiple and zero modes of deformations. Results are well presented and accompanied by analytical and finite element modeling techniques to characterize the structure's behavior. The experimental and theoretical methodology contained in this manuscript is excellent and the presented results provide useful insights on how this concept can be exploited to achieve mechanical signal guiding, buckling modes, and phonon spectra. However, shortcomings exist that must be addressed before publication. Please see the comments below.

1) The authors claim, "Here we introduce an all-in-one class of reprogrammable metamaterials that enable the in-situ reprogramming of zero modes to access the apparently conflicting properties of all classes", saying that they can reversibly switch between a structure that is floppy, multi-stable, and rigid. The work excellently demonstrates that the structure can switch between being floppy or rigid, where a different global response to a specific applied load is demonstrated. However, it lacks a demonstration of re-programmable multi-stability.

Reading this manuscript and observing the videos, it appears that the presented structure is always multi-stable and that the driving principle through which the structure can switch between a floppy and rigid state is driven by selectively snapping/actuating the metahinges of the structure. The configurations represented in figure 4e, f, and g can be classified as different stable states of the structure, each of which shows a different global mechanical property. Isn't a metahinge a bi-stable element as it undergoes fully reversible shape transformation between the two stable states and self-locks into the two configurations?

In existing literature, there are a few papers that have demonstrated tuning post-fabrication of the multi-stability of structure, where they demonstrate that the structure can reversibly switch between being monostable and multi-stable in situ. For example:

S. Li, K. W. Wang, Fluidic origami with embedded pressure dependent multi-stability: A plant inspired innovation, *Journal of the Royal Society Interface* 12 (111) (2015) 20150639. doi:10.1098/rsif.2015.0639 shows that the energy landscape of the origami structures changes with the pressure.

The authors are invited to revise their manuscript as it appears that the multi-stability of their structures is not reprogrammable but rather that the user can "unlock" different types of stable shapes by partially snapping the metahinges the structures.

2) Does partial actuation of structures with a rotating-square arrangement or a higher coordination number also show these different types of stable states? Can one generalize? Is not demonstrated that this property is valid for any geometry. This should be explained and clarified as it reads from the abstract and main text that this property is valid for any geometry.

3) Page 3, last paragraph it's written "For example, the yellow solid curve in Fig.1d exhibits a minor length change compared to the blue solid curve". "Minor" could be substituted with a more quantitative measure such as a difference in percentage.

4) At the end of page 4 the authors claim that they achieve a remarkable reduction of compression stiffness. How does this ratio compare to literature? For example, compared to papers [18-20] of their bibliography?

5) Figure 3b reports $N_0=2$, while Figure 3a, c, d, e report $N_0=1$. The definition of this variable should be reported in the capture or main text as it is unclear.

6) With which experimental method was the data reported in figure 3c,d,e captured? A note should be added to the methods.

7) In the methods page 15 authors write: "ensuring that the printed flexible hinges can repeatedly

undergo finite rotation without damage”, did the authors perform a cyclic test? In the main text, full reversibility is claimed, a specification of the number of cycles that the material can withstand would be needed. Or a strain analysis with a finite element model could prove that no plastic deformations occur?

Reviewer #2 (Remarks to the Author):

In this paper, authors present "an all-in-one class of reprogrammable metamaterials" that enables the in-situ reprogramming of zero modes to get properties of all classes using metahinges via different mechanisms. they demonstrate re-programmability for stiffness, mechanical signal guiding, buckling modes, phonon spectra, and auxeticity, opening a plethora of opportunities for all-in-one materials and devices.

The paper is well-written in overall and seems to be already been reviewed. It is acceptable for publication in the current state.

I would strongly recommend to the authors to cite some of the relevant work that have been published over the last 2 years on the reprogrammable metamaterials and those using contacts to change their properties.

Nature Communications 14, 4778, 2023
Composite Structures 319, 117151, 2023
Advanced Materials, 2210993 2023

Reviewer #3 (Remarks to the Author):

The authors demonstrate an ingenious mechanism to program the connectivity of a 2D tessellated lattice. Starting with one unit cell that is locally multistable, they proceed to change the "global" mechanics of the tessellated structure. I find the manuscript comprehensive and easy to follow, though I have several questions that I hope the authors can address.

1. It seems that programming requires a lot more force than global compression can provide, so the programming state cannot be erased. Does the same hold if the lattice experiences tension?
2. Can you discuss how such a structure can be abstracted as a "material"? Specifically, if we were to homogenize such a unit cell, is it possible to correlate the current state with a specific constitutive relation?
3. The unit cell level programming results in discontinuous changes in mechanical response. Is it possible to solve the inverse problem of "here's the global target behavior, which unit cell should I program to achieve that target behavior" through gradient descent or similar methods?
4. Barring practical limitations, can we scale this system up / down?
5. does the sequence of programming affect the mechanics? does it require more force to program some units than some other units? Are the edge units more floppy?

Response to Reviewer #1

This excellent manuscript includes the investigation of a metamaterial design that exploits the use of multi-stability to reversibly switch between a floppy mechanism and a stiff structure. A comprehensive experimental study is undertaken that uncovers how the concept can be employed to switch between multiple and zero modes of deformations. Results are well presented and accompanied by analytical and finite element modeling techniques to characterize the structure's behavior. The experimental and theoretical methodology contained in this manuscript is excellent and the presented results provide useful insights on how this concept can be exploited to achieve mechanical signal guiding, buckling modes, and phonon spectra. However, shortcomings exist that must be addressed before publication. Please see the comments below.

We are grateful for your appreciation and valuable comments.

1. The authors claim, “Here we introduce an all-in-one class of reprogrammable metamaterials that enable the in-situ reprogramming of zero modes to access the apparently conflicting properties of all classes”, saying that they can reversibly switch between a structure that is floppy, multi-stable, and rigid. The work excellently demonstrates that the structure can switch between being floppy or rigid, where a different global response to a specific applied load is demonstrated. However, it lacks a demonstration of re-programmable multi-stability.

Reading this manuscript and observing the videos, it appears that the presented structure is always multi-stable and that the driving principle through which the structure can switch between a floppy and rigid state is driven by selectively snapping/actuating the metahinges of the structure. The configurations represented in figure 4e, f, and g can be classified as different stable states of the structure, each of which shows a different global mechanical property. Isn't a metahinge a bi-stable element as it undergoes fully reversible shape transformation between the two stable states and self-locks into the two configurations? In existing literature, there are a few papers that have demonstrated tuning post-fabrication of the multi-stability of structure, where they demonstrate that the structure can reversibly switch between being monostable and multi-stable in situ. For example: S. Li, K. W. Wang, Fluidic origami with embedded pressure dependent multi-stability: A plant inspired innovation, *Journal of the Royal Society Interface* 12 (111) (2015) 20150639. doi:10.1098/rsif.2015.0639 shows that the energy landscape of the origami structures changes with the pressure.

The authors are invited to revise their manuscript as it appears that the multi-stability of their structures is not reprogrammable but rather that the user can “unlock” different types of stable shapes by partially snapping the metahinges the structures.

Thank you for your comments on this important matter. Multistability can exist at the sub-unit-cell level (local multistability showing no macroscopic strains) and at the unit-cell level (global multistability with macroscopic strains).

The “local bistability” of the metahinge is not reprogrammable, i.e., it is always active.

However, the “global multistability” of the Kagome-type metamaterial is reprogrammable, as shown in Figure R1, where a distinction between local and global is added at the bottom of each subfigure. Our Kagome metamaterial leverages the “local bistability” of metahinges to control the

“global multistability” of the periodic metamaterial. Our method for reprogramming multistability is distinct from those in the literature, including the one listed by the reviewer; existing works often rely on external means, such as pneumatic [1] or thermal [2, 3] loads. Our approach, however, enables us to intrinsically reprogram the internal architecture of the metamaterial.

Figure R1: Leveraging “local bistability” to reprogram “global multistability”.

Furthermore, a direct comparison between our current work and that by Rafsanjani [4] can further validate that our metamaterial has reprogrammable global multistability (see Figure R2). At the top of Figure R2a is our fully activated Kagome, analogous to the open state of the globally monostable kirigami illustrated in Figure R2b. Our selectively activated Kagome (bottom of Figure R2a) resembles the globally multistable kirigami illustrated in Figure R2c. As opposed to changing the pattern of cuts in a kirigami material [4], our metamaterial allows a selective deactivation of metahinges (adding the next nearest bonds to the Kagome) to activate the global multistability, a fully reversible reprogramming process without introducing any new perforations to the material.

Figure R2: **a**, Our Kagome metamaterial reprogrammed to be globally monostable (top) or multistable (bottom). **b**, Globally monostable kirigami with triangular cuts. **c**, Globally multistable kirigami by Rafsanjani [4].

In the original manuscript, the activated multistability of the Kagome-type metamaterial is demonstrated in two ways: 1) the non-zero energy barrier of the minimum energy path of the lattice

analogy, and 2) the multistable auxetic behavior of the physical sample shown in Supplementary Video 5.

To clarify this aspect, necessary changes have been added to the main text, also reported below.

*A fully deactivated Kagome metamaterial represents a stable structure, whereas a fully activated Kagome metamaterial is an unstable mechanism. A natural question arises: can a Kagome metamaterial with a selectively activated portion of its metahinges become metastable, i.e., **is it possible to reprogram the local state of each bistable metahinge to activate the global multistability of the metamaterial?***

2. Does partial actuation of structures with a rotating-square arrangement or a higher coordination number also show these different types of stable states? Can one generalize? Is not demonstrated that this property is valid for any geometry. This should be explained and clarified as it reads from the abstract and main text that this property is valid for any geometry.

Thank you for raising these meaningful questions.

To address them, we have now summarized all the material tessellations, i.e., metamaterial types, covered by this manuscript in Figure R3. The all-in-one characteristic is not valid for any geometry. Among the metamaterials we reported, only the Kagome-type metamaterial can transform among a rigid structure, a floppy mechanism, and a multistable matter (global). Our generalizations pertain to 1) the tessellation pattern of the metamaterial and 2) the metahinge layout as described below.

1) The rotation-square metamaterial is a demonstrative generalization of the tessellation of the C_2 metahinge (nodal connectivity $Z = 4$). Its characteristic, however, is that it can be activated or deactivated to behave as either a mechanism (fully activated) or a structure (fully or partially deactivated). Other generalized tessellations of the C_2 metahinge are also possible, and one instance with nodal connectivity $Z = 6$ is shown in Figure R4. The reprogrammable mechanical properties of these generalized tessellations require further exploration that is beyond the scope of the current manuscript.

2) The C_3 and C_4 metahinges are generalizations of the C_2 metahinge, but the isochoric characteristic no longer exists. The metamaterial comprising C_3 metahinges can be activated or deactivated to function as either a mechanism or a structure. The metamaterial comprising C_4 metahinges is always a floppy mechanism, no matter if it is activated or deactivated. A more generalized version of the metahinge with a higher order of cyclic symmetry is impractical because of the decaying spacing angle between adjacent rotation bodies, as explained at the end of the section entitled “Generalization to metamaterials comprising metahinges with a higher coordination number”.

To clarify that it is the Kagome-type metamaterial that can integrate all these three behaviors and that other metamaterials are generalization examples of either tessellation or metahinge design with reprogrammable zero modes, we made necessary revisions in the Abstract and the Outlook sections, which are highlighted in red.

	Structure	Reprogrammable global multistability	Mechanism
Kagome comprising C_2 metahinges			Rotation square comprising C_2 metahinges			Metamaterial comprising C_3 metahinges			Metamaterial comprising C_4 metahinges			
Figure R3: Summary of the metamaterials investigated in this manuscript.

Figure R4: Tessellation of the C_2 metahinge with nodal connectivity $Z = 6$.

3. Page 3, last paragraph it's written "For example, the yellow solid curve in Fig.1d exhibits a minor length change compared to the blue solid curve". "Minor" could be substituted with a more quantitative measure such as a difference in percentage.

Thank you for the suggestion.

The second local minimum points on the yellow and blue energy landscapes are at $\Delta L/L = 0.04$ and $\Delta L/L = 0.15$, respectively. The length change of the former is about 27% that of the latter. This quantitative description has been added to the main text, which is also reported below.

By tuning $\overline{OC}/\overline{AA'}$ and $\overline{A'B}/\overline{AA'}$, we can effectively alter the length change $L^ - L$ and eventually find conditions that are length preserving. In Fig. 1d, for instance, the yellow energy landscape exhibits $(L^* - L)/L = 0.04$, corresponding to a length change approximately 27% that observed in the blue energy landscape, where $(L^* - L)/L = 0.15$.*

4. At the end of page 4 the authors claim that they achieve a remarkable reduction of compression stiffness. How does this ratio compare to literature? For example, compared to papers [18-20] of their bibliography?

Thank you for the comment. Reference [19] by Wang [5] in the original manuscript focuses on reprogramming the bending stiffness of a metamaterial plate. References [18] by Chen [6] and [20] by Wu [7] are more relevant to the current manuscript since they investigated the reprogrammable compression stiffness of a bulk metamaterial. Below, we make a direct comparison between them.

Our fully activated Kagome metamaterial has an equivalent compression stiffness of 0.46 N/mm, showing a **79%** reduction compared to the value in its fully deactivated state (2.15 N/mm). The metamaterial chessboard reported by Chen [6] can achieve a stiffness reduction of up to **90%**, and the three-dimensional metamaterial by Wu [7] can promise a **72%** drop of stiffness prior to the snap-through buckling. With respect to the above two papers and other field-responsive mechanical metamaterials (about **38%** reduction) [8], it is fair to state that the stiffness reprogramming capacity of our Kagome metamaterial is remarkable.

A quantitative description is added to the main text. The revised text is reported below.

In a nutshell, from the fully deactivated state to the fully activated state, the emergence of multiple ZMs enables a remarkable reduction in compression stiffness of approximately 79%, and the deformation changes from a homogeneous bulk mode (affine) to a localized (non-affine) mode (Supplementary Video 2).

5. Figure 3b reports $N_0=2$, while Figure 3a, c, d, e report $N_0=1$. The definition of this variable should be reported in the capture or main text as it is unclear.

Thank you for the suggestion. N_0 refers to the number of zero modes of the lattice analogy, which has been defined in the last sentence of the section entitled “**Lattice analogy for isochoric re-configuration of Kagome metamaterials**” in the original manuscript. In the revised version, the meaning of N_0 has also been explained in the caption of Figure 3 as suggested by the reviewer.

$N_0=2$ in Figure 3b and $N_0=1$ in Figures 3a, c, d, and e are correct. Figure 3b shows the superposition of two independent zero modes, while Figures 3a, c, d, and e demonstrate the deterministic mechanical signal transmission in single-degree-of-freedom lattices.

6. With which experimental method was the data reported in figure 3c,d,e captured? A note should be added to the methods.

Thank you for the suggestion.

Equilateral triangular markers in red are attached to the center of each rotation body to trace the local rotation of the biaxial zero modes in the selectively activated Kagome metamaterial. The bottom-left corner of the finite-period specimen is clamped to freeze global rigid-body motions. A slow push is applied to the free edge of the rightmost rotation body to trigger the biaxial zero mode. The quasi-static deformation process is recorded by Sony RX 100 (Sony Corporation, Japan) with a sampling frequency of 60 Hz. Representative frames are extracted from the video at intervals of 1.67 seconds and imported into Microsoft Visio (Microsoft Corporation, United States). Masks are added to the red triangular markers of interest (see the outlined triangles in Fig. 3c, d, and e). The modal amplitudes along the \mathbf{e}_1 and \mathbf{e}_2 axes are then evaluated as the average rotation of the corresponding masks, denoted by $\bar{\theta}_{\mathbf{e}_1}$ and $\bar{\theta}_{\mathbf{e}_2}$ respectively.

The above description of the mechanical signal guiding through biaxial zero modes has been added to the Method section.

7. In the methods page 15 authors write: “ensuring that the printed flexible hinges can repeatedly undergo finite rotation without damage”, did the authors perform a cyclic test? In the main text, full reversibility is claimed, a specification of the number of cycles that the material can withstand would be needed. Or a strain analysis with a finite element model could prove that no plastic deformations occur?

Thank you for the suggestion.

The flexible hinges in our metamaterials undergo finite rotation mainly during activation and deactivation. A cycle of activation and deactivation of the metahinge is shown in Figure R5.

Figure R5: Activation and deactivation of the metahinge. Squeezing the hinges in the middle for activation and pulling the two boundaries for deactivation.

To demonstrate the repeatability and reversibility of our metamaterial, we repeated the process illustrated in Figure R5. At intervals of every 10 cycles, we tested its force-displacement response for activation. This set of data constitutes the experiment uncertainty regime, as plotted by the shaded area in Figure R6. The 1st activation and the 101st activation responses are plotted by the solid and dashed curves respectively. After 100 cycles of activation and deactivation, the limit force for snap-through changes from 13.1 N (1st activation) to 10.7 N (101st activation). This can be attributed to the plastic deformation developed within the flexible hinges, which were stabilized after a certain number of cycles [7, 9]. After 100 cycles, our metahinge can still promise a robust activated state, and there is no visible damage found in the flexible hinges, indicating that our metahinge can endure even more activation/deactivation cycles. The critical number of cycles and the optimization of the flexible hinge [10] to maximize this critical number are interesting topics that require further investigation and are beyond the scope of this paper.

The result above has now been updated in Fig. 1f, and the revised text is reported below.

Fig. 1f illustrates the force-displacement relation of the specimen that preserved its length upon activation. A pronounced snap-through instability appears under a pair of squeezing forces, followed by an abrupt and steep rise of the reaction force F_y at the onset of self-contact (purple spot on the solid blue curve in Fig. 1f). The reaction force F_y does not display a significant negative value mainly due to the viscoelasticity of the base material and the untethered loading condition, yet the activated state can be robustly preserved with self-contact. On the other hand, the metahinge is deactivated if a boundary-pulling force is applied to overcome the energy barrier and restore its original shape (Supplementary Video 1). To demonstrate the repeatability of the metahinge, we cyclically performed the activation and deactivation process. At intervals of every 10 cycles, we recorded the corresponding force-displacement relation for activation. This set of data is plotted as the experimental uncertainty domain shown in Fig. 1f. Plastic deformation may develop within the flexible hinges and stabilize after a certain number of cycles. The 101st activation exhibits a limit force of 10.7 N, a value that is approximately 82% that of the 1st activation. Despite this, the metahinge after cyclic usage can still promise a robust activated state, and the rotational ZM post-activation is well preserved.

Figure R6: Activation of the metahinge after a finite number of cycles.

The base material of our metamaterials is thermoplastic polyurethane (TPU). As we explained above, plastic deformation could occur within the flexible hinges but would get stabilized after a certain number of cycles [7, 9]. This stabilized plastic deformation does not affect the expected function of the metahinge as far as we have explored in this work.

Response to Reviewer #2

In this paper, authors present "an all-in-one class of reprogrammable metamaterials" that enables the in-situ reprogramming of zero modes to get properties of all classes using metahinges via different mechanisms. they demonstrate re-programmability for stiffness, mechanical signal guiding, buckling modes, phonon spectra, and auxeticity, opening a plethora of opportunities for all-in-one materials and devices.

The paper is well-written in overall and seems to be already been reviewed. It is acceptable for publication in the current state.

We are sincerely grateful for your feedback.

I would strongly recommend to the authors to cite some of the relevant work that have been published over the last 2 years on the reprogrammable metamaterials and those using contacts to change their properties.

Nature Communications 14, 4778, 2023

Composite Structures 319, 117151, 2023

Advanced Materials, 2210993 2023

Thank you for the suggestion and for sharing these interesting papers with us.

In the revised Introduction, two recent publications focusing on field-responsive [11] and reconfigurable [12] mechanical metamaterials have been supplemented to contextualize our work within the latest progress in the field of reprogrammable metamaterials. The corresponding sentence in the Introduction is reported below.

To enable on-the-fly adjustment of stiffness or stability, reprogrammable mechanical metamaterials have been rationally designed to incorporate field-responsive constituents^{15, 16, 17} or reconfigurable architecture^{1, 18, 19, 20, 21, 22}.

Self-contact between elements of the internal architecture is a prevalent phenomenon observed in rotation-based metamaterials, particularly when subjected to finite deformation conditions [13, 14, 15]. Self-contact always indicates a change in the domain connectivity, i.e. a topological change in the physical space, and hence it is useful for substantially reprogramming the effective characteristics such as stiffness [6] and snap-through buckling [7] of a metamaterial. In literature, however, the self-contact of most rotation-based metamaterials is only a transient behavior emerging during large deformation [13, 14, 15]; self-contact disappears upon the removal of the applied load. In contrast, the self-contact in our metamaterials can be robustly preserved through the local bistability of the metahinge, making it a stable property of the activated material without any reliance on continuous stimulus supply. To address the reviewer's suggestion, the two most relevant publications [13, 14] have been added at an appropriate position in the main text, where we first introduce the concept of self-contact. The corresponding sentence in the manuscript is also reported below.

Under the squeeze of a pair of activation forces (red arrows in Fig. 1b), the architecture can undergo moderate local rotation, making the pairing edges (purple) and hinges (green) get close and eventually merge due to self-contact ^{28, 29}.

Response to Reviewer #3

The authors demonstrate an ingenious mechanism to program the connectivity of a 2D tessellated lattice. Starting with one unit cell that is locally multistable, they proceed to change the "global" mechanics of the tessellated structure. I find the manuscript comprehensive and easy to follow, though I have several questions that I hope the authors can address.

We appreciate your insightful feedback and have addressed your comments point-by-point below.

1. It seems that programming requires a lot more force than global compression can provide, so the programming state cannot be erased. Does the same hold if the lattice experiences tension?

Thank you for the comment.

The "programming" (activated) state of our metamaterials can be erased (deactivated), i.e., being reset to the stress-free state. The activation/deactivation of the metahinge has been demonstrated in Supplementary Video 1, which is also illustrated in Figure R5. The activation/deactivation of the Kagome metamaterial has been demonstrated in Supplementary Video 2.

Global biaxial tension can be leveraged to deactivate an activated metamaterial, but the efficiency is dependent on the tessellation type of the metahinge. The fully activated Kagome metamaterial, for example, necessitates an internal force field, as indicated by the red arrows on the left side of Figure R7, for deactivation. An applied global biaxial tension (indicated by the blue arrows), however, cannot fully trigger the required internal force field, meaning that some of the metahinges might remain in the activated states unless a larger tension force is applied at the boundaries. In the case of the fully activated rotation-square metamaterial (right side of Figure R7), the global biaxial tension aligns with the orientation of the metahinges. Therefore, the entire rotation-square metamaterial can be fully deactivated from its activated state simply through global biaxial tension.

Despite that global tension may offer a faster route for deactivation, we deem that harnessing local perturbations for the deactivation of each individual metahinge provides a more controllable approach, as demonstrated in Supplementary Video 2.

Figure R7: Required internal force for deactivating a fully activated metamaterial and illustration of global biaxial tension.

2. Can you discuss how such a structure can be abstracted as a "material"? Specifically, if we were to homogenize such a unit cell, is it possible to correlate the current state with a specific constitutive relation?

Thank you for the comments.

Our fully activated/deactivated metamaterials exhibiting a periodic tessellation can be treated as a homogeneous material. However, when selectively activated or deactivated, they become inhomogeneous materials. Additionally, leveraging a selectively activated metamaterial as a supercell to create a periodic tessellation can also yield a macroscopically homogeneous material, as demonstrated in Fig. 4h, i, and j.

Homogenization is possible for the linear mechanical properties of our activated or deactivated metamaterials. This can be implemented by combining the Asymptotic Homogenization Method and the Finite Element Method for a geometry-detailed model under periodic boundary conditions [16]. The lattice analogy with rationally chosen stiffness of bonds can also serve for homogenization. The phonon spectra (Fig. 4h, i, and j) derived from the Bloch theorem or the Floquet Periodic Boundary Condition can also be used to homogenize the lattice analogy [17], which can reflect the effective stiffness and anisotropy of the metamaterial in the low-frequency regime. In particular, the fully activated Kagome metamaterial and the fully activated rotation-square metamaterial can be homogenized as an isotropic and orthotropic linear elastic material respectively. Moreover, viscoelasticity can also be incorporated into the constitutive model to respect the nature of the thermoplastic base material, which is, however, beyond the scope of the current work.

Homogenization is inappropriate for describing the mechanics during activation. During activation, the local architecture of a unit cell undergoes significant local rotation driven by a pair of local forces, where there is no obvious macroscopic strain (the metahinge is designed to be isochoric to avoid geometry frustration). The homogenization method only accounts for forces exerted on the boundary of the unit cell but ignores the local features at the sub-unit-cell level. As

a result, we deem that the mechanics during activation may not be described by a homogenized model characterized by a conventional constitutive relation.

3. The unit cell level programming results in discontinuous changes in mechanical response. Is it possible to solve the inverse problem of "here's the global target behavior, which unit cell should I program to achieve that target behavior" though gradient descent or similar methods?

Thank you for the insightful comment.

It is possible to inversely design the activation pattern in a metamaterial for a given global target response if there are sufficient unit cells. The attainable response space is bounded between the responses of the fully activated (lower bound) and deactivated (upper bound) metamaterials (see Figure R8).

Figure R8: Attainable response space of an inverse problem (a partially activated metamaterial).

This inverse problem can be stated as

$$\begin{aligned} \min_{s_i} \int_0^{u_y} w(F_y^{\text{target}} - F_y) du_y \\ \text{s.t. } \mathbf{R}(s_i, u_y, F_y) = \mathbf{0} \\ s_i = 0 \text{ or } 1 \end{aligned}$$

where s_i is the design variable (the activated state of each metahinge), w is the weight function, F_y^{target} is the target force-displacement relation, F_y is the evaluated reaction force, u_y is the applied global displacement, and \mathbf{R} is the governing equation.

For given target response F_y^{target} and weight function w , this discrete optimization problem may manifest multiple optimal solutions. With these insights, we deem that a derivative-free method such as the Genetic Algorithm or the Particle Swarm Algorithm can be a more appropriate candidate.

4. Barring practical limitations, can we scale this system up/down?

Our theoretical analysis is rooted in geometric analysis and macroscopic solid mechanics. Consequently, our metamaterials offer scalability within a range spanning from millimeters to meters.

The minimum characteristic length of our metamaterial is the dimension of the living hinge, showing a width of 0.6 mm. This size may allow appropriate reduction when using the high-resolution Fused Deposition Modelling (FDM) or the Liquid Crystal Display (LCD) manufacturing method. As this size scales down, imperfections would have a more prominent impact on the durability of the living hinges and also the internal contact between rotation bodies, due to which the robustness of the activated state could be compromised.

Appropriate scaling up is also possible in our metamaterials. It should be pointed out that the out-of-plane thickness should also be scaled accordingly to prevent out-of-plane deformation of the metahinge during activation, a phenomenon that could make the activated state less robust.

A statement about the scalability of our metamaterials has been added to the Outlook section, which is also reported below.

Finally, we envision the scalability of our metahinge within a range spanning from millimeters to meters and the possibility of embedding the metahinge architecture into origami-type metamaterials, enabling on-demand activation of foldability, thus extending our concept from two-dimensional planar materials to three-dimensional bulk materials.

5. does the sequence of programming affect the mechanics? does it require more force to program some units than some other units? Are the edge units more floppy?

Thank you for the insightful comments.

Unlike other frustrated metamaterials whose final state is heavily dependent on the loading sequence/history [18], our metamaterial in its activated state is frustration-free (the metahinge is designed to be isochoric). As a result, the sequence of programming/activation would not affect the stiffness or stability of the activated metamaterial. A statement of the sequence-independent characteristic of our isochoric metamaterial is added to the main text, which is also reported below.

Upon a sequence of local activation forces (Supplementary Video 2), the honeycomb structure undergoes an isochoric reconfiguration where all the metahinges are activated. We point out that the activation sequence has a negligible influence on the resulting activated state due to the isochoric characteristic of each metahinge, and hence any consecutive actions of activation are independent of each other.

Despite there being no geometry frustration in the activated state, slight local incompatibility between adjacent metahinges can arise during activation. The metahinges residing at the free boundaries suffer less constraint from their neighbors, and hence they require less activation force compared to those metahinges in the bulk of the metamaterial.

Through selective activation of a finite-period specimen, we can produce floppy edge modes such as the one shown in Figure R9. These edge modes emerge from the removal of a specific group of the next nearest neighbor bonds from the lattice analogy, as opposed to the polarized edge modes that are obtained by breaking the unit cell symmetry of a regular Kagome lattice [19].

Figure R9: Selective activation of the Kagome metamaterial allowing the emergence of floppy edge modes.

References

- [1] Suyi Li and KW Wang. Fluidic origami with embedded pressure dependent multi-stability: a plant inspired innovation. *Journal of The Royal Society Interface*, 12(111):20150639, 2015.
- [2] Yunlan Zhang, Mirian Velay-Lizancos, David Restrepo, Nilesh D Mankame, and Pablo D Zavattieri. Architected material analogs for shape memory alloys. *Matter*, 4(6):1990–2012, 2021.
- [3] Hang Yang, Nicholas D’Ambrosio, Peiyong Liu, Damiano Pasini, and Li Ma. Shape memory mechanical metamaterials. *Materials Today*, 66:36–49, 2023.
- [4] Ahmad Rafsanjani and Damiano Pasini. Bistable auxetic mechanical metamaterials inspired by ancient geometric motifs. *Extreme Mechanics Letters*, 9:291–296, 2016.
- [5] Yifan Wang, Liuchi Li, Douglas Hofmann, José E Andrade, and Chiara Daraio. Structured fabrics with tunable mechanical properties. *Nature*, 596(7871):238–243, 2021.
- [6] Tian Chen, Mark Pauly, and Pedro M Reis. A reprogrammable mechanical metamaterial with stable memory. *Nature*, 589(7842):386–390, 2021.
- [7] Lei Wu and Damiano Pasini. In-situ activation of snap-through instability in multi-response metamaterials through multistable topological transformation. *Advanced Materials*, page 2301109, 2023.
- [8] Julie A Jackson, Mark C Messner, Nikola A Dudukovic, William L Smith, Logan Bekker, Bryan Moran, Alexandra M Golobic, Andrew J Pascall, Eric B Duoss, Kenneth J Loh, et al. Field responsive mechanical metamaterials. *Science Advances*, 4(12):eaau6419, 2018.
- [9] Matteo Gavazzoni, Stefano Foletti, and Damiano Pasini. Cyclic response of 3d printed metamaterials with soft cellular architecture: The interplay between as-built defects, material and geometric non-linearity. *Journal of the Mechanics and Physics of Solids*, 158:104688, 2022.
- [10] Xiao Shang, Lu Liu, Ahmad Rafsanjani, and Damiano Pasini. Durable bistable auxetics made of rigid solids. *Journal of Materials Research*, 33(3):300–308, 2018.
- [11] Bihui Zou, Zihe Liang, Dijia Zhong, Zhiming Cui, Kai Xiao, Shuang Shao, and Jaehyung Ju. Magneto-thermomechanically reprogrammable mechanical metamaterials. *Advanced Materials*, 35(8):2207349, 2023.
- [12] Xinyu Hu, Ting Tan, Benlong Wang, and Zhimiao Yan. A reprogrammable mechanical metamaterial with origami functional-group transformation and ring reconfiguration. *Nature Communications*, 14(1):6709, 2023.
- [13] Bolei Deng, Ahmad Zareei, Xiaoxiao Ding, James C Weaver, Chris H Rycroft, and Katia Bertoldi. Inverse design of mechanical metamaterials with target nonlinear response via a neural accelerated evolution strategy. *Advanced Materials*, 34(41):2206238, 2022.
- [14] Krzysztof K Dudek, Julio A Iglesias Martínez, Gwenn Ulliac, Laurent Hirsinger, Lianchao Wang, Vincent Laude, and Muamer Kadic. Micro-scale mechanical metamaterial with a controllable transition in the poisson’s ratio and band gap formation. *Advanced Materials*, page 2210993, 2023.

- [15] KK Dudek, L Mizzi, JA Iglesias Martínez, A Spaggiari, G Ulliac, R Gatt, JN Grima, V Laude, and M Kadic. Micro-scale graded mechanical metamaterials exhibiting versatile poisson's ratio. *Composite Structures*, 319:117151, 2023.
- [16] Sajad Arabnejad and Damiano Pasini. Mechanical properties of lattice materials via asymptotic homogenization and comparison with alternative homogenization methods. *International Journal of Mechanical Sciences*, 77:249–262, 2013.
- [17] Robert G Hutchinson and Norman A Fleck. The structural performance of the periodic truss. *Journal of the Mechanics and Physics of Solids*, 54(4):756–782, 2006.
- [18] Xiaofei Guo, Marcelo Guzmán, David Carpentier, Denis Bartolo, and Corentin Coulais. Non-orientable order and non-commutative response in frustrated metamaterials. *Nature*, 618(7965):506–512, 2023.
- [19] Charles L Kane and Tom C Lubensky. Topological boundary modes in isostatic lattices. *Nature Physics*, 10(1):39–45, 2014.

REVIEWERS' COMMENTS

Reviewer #1 (Remarks to the Author):

The reviewer thanks the authors for their diligent efforts to improve the manuscript and address the comments. The authors' responses and edits provided effective clarification and explanation for all my comments. The manuscript is ready for publication, provided that the authors address the two minor comments listed below:

1. The authors write: is it possible to reprogram the local state of each bistable metahinge to activate the global multistability of the metamaterial? The use of the word REPROGRAM appears inappropriate as the bistability of the metahinge is programmed before fabrication. The global multistability can be reprogrammed. It would be clearer for the readers if the author could state what they have written in the response letter, that is: is it possible to leverage "local bistability" to reprogram "global multistability".

2. Concerning their response to question 2, I suggest that the observations that have been written in their response letter are summarized and integrated into the section "Generalization to metamaterials comprising metahinges with a higher coordination number" as right now it is not clear to the reader the limit of the generalization while reading in this section.

Reviewer #3 (Remarks to the Author):

Thank you for addressing the comments. I recommend the publication of this manuscript.

Response to Reviewer #1

The reviewer thanks the authors for their diligent efforts to improve the manuscript and address the comments. The authors' responses and edits provided effective clarification and explanation for all my comments. The manuscript is ready for publication, provided that the authors address the two minor comments listed below:

1. The authors write: is it possible to reprogram the local state of each bistable metahinge to activate the global multistability of the metamaterial? The use of the word REPROGRAM appears inappropriate as the bistability of the metahinge is programmed before fabrication. The global multistability can be reprogrammed. It would be clearer for the readers if the author could state what they have written in the response letter, that is: is it possible to leverage “local bistability” to reprogram “global multistability”.

Thank you for requesting further clarification.

We would like to explain the following.

The **local state** of the metahinge can be reprogrammed to be either activated or deactivated. The **intrinsic bistability** of the metahinge is not reprogrammable. Our sentence reports that the local state can be reprogrammed, and hence it is correct. It appropriately conveys the intended message.

Yet given the reviewer's comment and to possibly avoid a misunderstanding, we amended the use of the term “reprogram”. The revised version is here below, where the changes are highlighted in red. In a nutshell, we replace “reprogram” with “switch”, but their meaning in this context is identical.

*is it possible to **selectively switch** the local state of each bistable metahinge to activate the global multistability of the metamaterial?*

2. Concerning their response to question 2, I suggest that the observations that have been written in their response letter are summarized and integrated into the section “Generalization to metamaterials comprising metahinges with a higher coordination number” as right now it is not clear to the reader the limit of the generalization while reading in this section.

Thank you for the suggestion. A statement is supplemented at the end of the section “Generalization to metamaterials comprising metahinges with a higher coordination number”, which is also reported below (changes are highlighted in red).

*We remark that the spacing angle τ in the fully activated state (Fig. 6a and e) is a crucial geometry parameter preventing us from reaching higher-order cyclic symmetry; τ decays exponentially with the symmetric order, and hence the internal contact between rotational bodies would disrupt the bistable activation process. **While the metamaterial tessellations incorporating C_3 and C_4 metahinges (Fig. 6) enable a reprogrammable number of ZMs, facilitating in-situ stiffness tuning, their global response lacks the ability to transition between being multistable and monostable, unlike the Kagome metamaterial depicted in Fig. 4e, f, and g.***